# Weathering of Roofing Insulation Materials under Multi-Field Coupling Conditions

**DOI:** 10.3390/ma12203348

**Published:** 2019-10-14

**Authors:** Shuangxi Zhou, Yang Ding, Zhongping Wang, Jingliang Dong, Anming She, Yongqi Wei, Ruguang Li

**Affiliations:** 1School of Civil Engineering and Architecture, East China Jiaotong University, Nanchang 330013, China; green.55@163.com (S.Z.); dongjingliang@ecjtu.edu.cn (J.D.); 2Department of Civil Engineering, Zhejiang University, Hangzhou 310058, China; 3School of Materials Science and Engineering, Tongji University, Shanghai 201804, China; wangzpk@tongji.edu.cn (Z.W.); sheanming@tongji.edu.cn (A.S.); wei_yongqi@tongji.edu.cn (Y.W.)

**Keywords:** thermal insulation material, thermal conductivity, weathering test, humid–heat–solid coupling, waterproof layer

## Abstract

Rigid polyurethane foam, foam concrete, and vacuum insulation board are common roofing insulation materials. Their weathering performance under long-term multi-field coupling determines the overall service life of the roof. The weathering properties of rigid polyurethane foam, foam concrete and vacuum insulation panels were studied under freeze thaw, humid-heat, dry-wet, high-low temperature, and multi-field coupling cycles, respectively. The heat transfer and construction process of roof panels was simulated base on upper loading and moisture transfer factors. The result indicates that the mass loss of the foam concrete and the rigid polyurethane foam in the weathering test was significant, which led to the gradual increase of thermal conductivity. Meanwhile, the thermal conductivity and mass loss of vacuum insulation panels did not change due to the lack of penetration under external pressure, therefore, it is necessary to construct composite thermal–insulation materials to alleviate the adverse effects of the service environment on a single material and realize the complementary advantages and disadvantages of the two materials. The results of the numerical simulations indicated that the roof structure must be waterproofed, and its weatherproof performance index should be the same as that of the thermal insulation material. Considering structural deformation, the overall heat transfer performance of the product was increased by around 5%.

## 1. Introduction

A comfortable thermal environment in a building has gradually become a necessity; however, building energy consumption has increased. Although a mixed design of insulation materials can reduce energy consumption, many new insulation materials have been proposed due to the environment of the roofing insulation system being complicated. The weathering resistance of insulation materials is a key indicator that directly affects the energy efficiency of the insulation system. Therefore, investigating the weathering resistance of insulation materials is necessary to ensure the service life of insulation materials is as long as that of the structure to which they are attached.

Thermal conductivity is an inherent material property, and the rate of mass loss can directly reflect the degree of damage to the material under different environmental conditions. Therefore, the thermal conductivity and the rate of mass loss can be determined using the macroscopic indices of the weathering resistance of thermal insulation materials [1,2,3,4,5]. Incorporating different proportions of mineral admixtures in insulation materials and then analyzing the influence of microscopic changes on the thermal conductivity of the materials using scanning electron microscopy is common, and based on this information, a new composite insulating material was developed [6,7,8,9,10]. Kalapathy et al. [11] conducted cyclic freeze–thaw testing of insulation materials, and measured changes in the thermal conductivity of such materials. Researchers have established heat transfer models for different materials and analyzed various factors affecting the heat transfer path. The existing studies focus on the durability of insulation materials in a single environment, but there are few weathering resistance studies on the thermal performance of insulation materials. Previous experimental studies have verified that insulation materials can meet the requirement of thermal conductivity under certain experimental conditions; however, when the composite material was prepared for a specific structure, the effect, after construction, was unsatisfactory. Due to the changing service environment and complicated structure of insulation materials, the new thermal insulation materials obtained with different mixing ratios may not meet weathering resistance requirements, so it is necessary to test the evolution of thermal conductivity of insulation materials under different environmental conditions.

Through applying computer simulation technology to simulate the evolution of insulation materials in complex environments, a qualitative judgment of the performance of insulation materials was researched [12,13,14]. Some researchers studied the characteristics of porous media, establishing a heat transfer equation to clarify the influence of several factors, such as humidity and density, on the heat transfer process [15,16,17,18]. However, owing to the influence of the thermal stress load and human-induced load, and their effects on the roofing structure, it is necessary to consider the influence of these loads on the heat transfer process in any multi-field coupling model, and combine various influential factors to obtain the evolution of macro-performance for the whole roofing insulation system.

Herein, thermal conductivity and mass loss rate were applied as macroscopic indicators. The weathering resistance tests under a freezing–thawing condition, humid–heat condition, dry–wet environment, high–low temperature, and multi-field coupling cycles were carried out on three typical insulation materials: rigid polyurethane foam, foam concrete, and vacuum insulation panels. Combining the data with mass transfer theory for porous media, the effects of load and humidity on heat transfer were qualitatively determined. The multi-field coupling model was established and compared with experimental data. Material optimization design and structural design for the actual roofing insulation structure were conducted using the multi-field coupling equations, and a numerical simulation was conducted for the thermal bridge scheme.

## 2. Materials and Methods

### 2.1. Test Materials

A 300 mm × 300 mm × 30 mm rigid polyurethane foam (RPUF), produced by Shanghai Huafeng Rigid Polyurethane Foam Co., Ltd. (Shanghai, China); a 300 mm × 300 mm × 15 mm vacuum insulation panel (VIP), which was made of nanoporous core material, produced by Qingdao Kerui (Qingdao, China); a 300 mm × 300 mm × 25 mm foam concrete (FC) produced by Shanghai Shutong Building Materials Co., Ltd. (Shanghai, China); and a binding mortar produced by Shanghai Shun’an (Shanghai, China) were selected. The composite roofing materials were obtained using combining rigid polyurethane foam, vacuum insulation panels, and foam concrete in pairs by using binding mortar. They are shown in Table 1.

### 2.2. Test Instrument

In this paper, the thermal conductivity was tested using the single plate thermal conductivity test instrument, which can be seen in Figure 1. The instrument model was IMDRY300-II, which was produced by Tianjin Yingbei Technology Development Co., Ltd. (Tianjin, China). The determination of thermal conductivity was based on the GB/T 13475-2008 “Calibration of Adiabatic Steady State Heat Transfer Property and Protection of Hot Box Method.”

### 2.3. Test Method

Since the three roofing materials were organic, inorganic, and new materials separately, national standards and local codes provide no unified classification of their weathering resistance. Therefore, the experimental design was carried out with reference to the similar standards, and the weathering resistance of the three insulation materials was obtained.

#### 2.3.1. Freeze–Thaw Cycle Test

The freeze–thaw cycle test was carried out according to the GB/T 50082-2009 Standard for Test Methods of Long-Term Performance and Durability of Ordinary Concrete. The sample was immersed in water at a temperature of 20 ± 2 °C for 4 days in the early stage of the test. The water level was 30 mm above the upper surface of the sample during the immersion. The freeze–thaw cycle test was then carried out: The freezer temperature was adjusted to between −18 and −20 °C, and the freezing time of the samples was 4 h. Immediately after the end of the freezing, the samples were transferred to a water tank at a temperature of 18–20 °C to allow them to thaw, where the water level in the water tank was 30 mm above the upper surface of the samples, and the thawing time was 4 h, which was recorded as a freeze–thaw cycle.

#### 2.3.2. Humid–Heat Cycle Test

The humid–heat environment was constructed according to the GB/T 12000-2003 Determination of the Effects of Exposure to Damp Heat, Water Spray, and Salt Mist for Plastics. The temperature in the constant temperature and humidity chamber was set to 60 °C and the relative humidity (RH) was 93%. The test durations were 7, 14, 28, 56, and 112 days. Three replicates were established for each test duration; the thermal conductivity and mean average mass of the three samples were taken as representing the measured results.

#### 2.3.3. Dry–Wet Cycle Test

The dry–wet cycle test was carried out in accordance with the provisions of the dry–wet cycle test in the GB/T 11969 2008 Test Methods of Autoclaved Aerated Concrete. First, each sample was baked in a fan-assisted electro-thermal drying oven at 60 °C to a constant mass; the samples, in groups of three, were then cooled at a room temperature of 20 ± 5 °C for 20 min, and then placed in a water tank. The samples were immersed in water at a temperature of 20 ± 5 °C; the water level was maintained at 30 mm above the upper surface of the sample, and the samples were removed after 5 min, dried for 30 min, and then put into the oven (heated to 60 °C), and the samples were baked for 7 h. That is, the samples were baked at 60 °C for 7 h, cooled for 20 min, and placed in water at 20 ± 5 °C for 5 min, representing the complete dry–wet cycle. For it to be a complete cycle, the sample need to return to being baked. 

#### 2.3.4. High–Low Temperature Cycling

The high–low temperature cycling test was carried out in accordance with the GB/T 2423.34-2012 Environmental Testing Part 2—Test Methods ZAD: Composite Temperature/Humidity. Before the test, each sample was oven-baked at 60 °C to a constant mass, and then the samples were placed in an incubator, and the temperature was kept at 60 °C for 3h then the temperature was lowered to −20°C for 3h. During the test, the humidity of the incubator was set to 50% RH, except when the temperature was below zero. That is, the samples were exposed to high temperature at 60 °C and exposed to low temperature at −20 °C for 3 h, respectively. This was the high–low temperature cycle. For it to be a complete cycle, the sample need to return to being baked. 

#### 2.3.5. Multi-Field Coupling Cycle

The temperature and humidity control was carried out in accordance with the weathering test method for thermal insulation materials in the industry standard JGJ 144-2004 Technical Specification for Interior Thermal Insulation on External Walls. The test method was as follows: (1) Samples were heated and humidified for 8 h. The surface temperature of the samples was adjusted to 60 °C over a 1-h period, the humidity was adjusted to 93% RH, and the temperature and humidity were kept constant for 7 h in this state. (2) The samples were then cooled by lowering the surface temperature of the samples from 60 °C to −20 °C over 2 h, and the samples were then maintained in this state for 14 h. The surface temperature and humidity changes of the samples are shown in Figure 2. A cycle was completed within 24 h over test durations of 7, 14, 28, 56, 84, and 112 days. During the test, the pressure was kept constant (under self-weight conditions): 1 kg of 300 mm × 300 mm cement board and 0.5 kg fine sand was laid on the thermal insulation material, and then the cement board was placed on the sand ensuring that the surface of the fine sand was flat. During testing, the stress field was kept constant. For multi-field coupling, the three fields of temperature, humidity, and stress were considered, in which the temperature field and the humidity field alternated over a cycle of 24 h, and the load remained unchanged at 15 N. The multi-field coupling of the structure fully considered the factors that were unfavourable to the thermal performance of the roofing insulation material, such as high–low temperature circulation, freeze–thaw circulation, and humid–heat aging. The established range of field strengths covered the service environments of most roofing systems encountered in China.

### 2.4. Simulation Method

COMSOL Multiphysics^®^ is a general-purpose simulation software for modeling designs, devices, and processes in all fields of engineering, manufacturing, and scientific research. The platform product can be used on its own or expanded with functionality from any combination of add-on modules for structural mechanics, heat transfer, et al. [19]. Therefore, COMSOL three-dimensional simulation software was used to conduct thermal conduction, humid–heat coupling (considering or not considering the effect of a waterproof layer), and qualitative calculations of heat–solid coupling in this paper [20,21,22].

Considering the summer extreme climate in the hot summer and cold winter regions, the roof temperature was set to 60 °C and the relative humidity was 0.6, the indoor temperature was set to 20 °C, the relative humidity was 0.3, and the convective heat transfer coefficient was assumed to be 5 W·m^−2^·K^−1^. When considering the effect of the waterproof layer and not considering the drainage, that is, the liquid water did not diffuse inwards, and assuming that there was a 2-mm water layer at the top of the roof. When the effect of the waterproof layer was not considered, the liquid water would diffuse inwards. The permeability of the materials was input as a model parameter, and humid–heat coupling simulations were performed. Regardless of the pressure during construction and the damage to the roof and its component parts, the applied uniform distributed load was assumed to be 2.0 (kN/m^2^) based on the load specification.

#### 2.4.1. Heat Transfer Model

In order to obtain a good heat transfer model, the theory base on the following equation:(1)$$\frac{\partial }{\partial t}c_{p,m}\rho_{m}T\;+\;c_{p,m}\rho_{m}\nabla T\;=\;\nabla \lambda \nabla T\;+\;Q$$
where λ  is the thermal conductivity (W·m^−1^·K^−1^), the negative sign on the right-hand side of the equation indicates that the direction of the heat flux is opposite to the direction of the temperature gradient, $\rho_{m}\;$ is the density of the dry material (kg/m^3^), $c_{p,m}\;$ is the specific heat of the dry material (J·kg^−1^·K^−1^), and *Q* is the heat source (W/m^3^).

#### 2.4.2. Humidity Transfer Model

Since the roofing material was an isotropic continuous porous medium, humidity transfer was expressed using Equation (2) according to mass conservation in the unit body:(2)$$\frac{\partial \omega }{\partial t}\;=\;-\nabla \left(j_{l}\;+\;j_{v}\right)$$
where ω  is the volumetric moisture content (kg/m^3^), *t* is the time (s), $j_{v}\;$ is the water vapor transmission rate (kg/m^2^s); and $j_{l}\;$ is the liquid water transfer rate (kg/m^2^s).

#### 2.4.3. Humid–Heat Coupling Model

The volumetric moisture content was taken as a function of the material moisture content *U* and temperature *T*, as shown in Equation (3):(3)ω = f(U,T)

Equation (4) was obtained via derivation of both sides with respect to time *t*:(4)$$\frac{\partial \omega }{\partial t}\;=\;\frac{\partial \omega }{\partial U}\cdot \frac{\partial U}{\partial t}\;+\;\frac{\partial \omega }{\partial T}\cdot \frac{\partial T}{\partial t}$$

#### 2.4.4. Solid–Heat Coupling

When the temperature of the elastomer changes, it will expand or contract as the temperature increases or decreases; if the elastomer is unconstrained and its expansion or contraction can occur freely, no stress is generated in the elastomer. However, when the external constraint to the elastomer or the mutual constraint among various parts of the elastomer causes such an expansion or contraction to be obstructed, thermal stress is generated therein:(5)$$\sigma_{i,j}\;=\;2G\varepsilon_{i,j}\;+\;\left(A_{e}\;-\;\beta_{T}\right)\delta_{i,j}$$

The equilibrium equation is shown as Equation (6):(6)$$\nabla^{2}u_{i}\;+\;\frac{1}{1\;-\;2\mu }\frac{\partial \varepsilon_{kk}}{\partial i}\;-\;\frac{2\left(1\;+\;\mu \right)}{1\;-\;2\mu }\frac{\partial T}{\partial i}\;+\;\frac{f_{i}}{G}\;=\;0$$

The compatibility equation is shown as Equation (7):(7)$$\nabla^{2}\delta_{i}\;+\;\frac{1}{1+\mu }\frac{\partial^{2}\delta_{kk}}{\partial i^{2}}\;=\;-\alpha_{E}\left(\frac{1}{1+\mu }\nabla^{2}T\;+\;\frac{1}{1+\mu }\frac{\partial^{2}T}{\partial i^{2}}\right)\;-\;\left(\frac{\mu }{1-\mu }\frac{\partial f_{i}}{\partial i}\;+\;2\frac{\partial f_{i}}{\partial i}\right)$$
where, $\beta \;=\;\frac{\alpha_{E}}{1-2\mu }\;=\;\lambda \alpha_{E}\;+\;2G$, $G\;=\;\frac{E}{2\left(1+\mu \right)}$, $\;A_{e}=\;\frac{\mu E}{\left(1-2\mu \right)\left(1+\mu \right)}$, $\delta_{kk}\;=\;\delta_{xx}\;+\;\delta_{yy}\;+\;\delta_{zz}$, $\varepsilon_{kk}\;=\;\varepsilon_{xx}\;+\;\varepsilon_{yy}\;+\;\varepsilon_{zz}$, E is the elastic modulus, $\alpha_{E}\;$ is the coefficient of linear thermal expansion,  μ is the Poisson ratio, and $f_{i}$ is the component of the unit volume force on the coordinate axis.

## 3. Results and Discussion

### 3.1. Test Results and Discussion

#### 3.1.1. Thermal Conductivity of Rigid Polyurethane Foam

As shown in Figure 3a, in the humidity–heat aging test, the thermal conductivity of the rigid polyurethane foam changed at day 14, and in the subsequent three tests, the thermal conductivity increased 3%, 8%, and 10%, and reached 29.98% in 112 days. In the dry–humidity cycle, the thermal conductivity of the rigid polyurethane foam increased from the initial value of 0.0150 W·m^−1^·K^−1^ to 0.0200 W·m^−1^·K^−1^. Due to the pore structure of the rigid polyurethane foam changing during these two weathering resistance tests, and with increased aging time, the rigid polyurethane foam pore structure was larger, which can be seen in Figure 4. Therefore, the heat transfer path of the heat flow was shortened and the thermal conductivity of the rigid polyurethane foam increased [23].

In the high–low temperature test, the thermal conductivity of the rigid polyurethane foam increased by 16.65% after 112 high–low temperature cycles, and the rate of mass loss was only 1.63% (Figure 3b). It can be inferred that the diffusion coefficient of the foaming agent gas in the closed pores of the matrix increased in this high-temperature environment [24]. In the high–low temperature cycle test, the temperature rise caused an increased gaseous exchange between the foaming agent and the air, resulting in an increase in the thermal conductivity of the rigid polyurethane foam.

In the freeze–thaw cycle test, the thermal conductivity of the rigid polyurethane foam increased by 33.12% after 112 days; this was because the liquid water expanded when frozen and caused the pore walls to crack under tension. Due to the high closed pore ratio of the rigid polyurethane foam, this was mainly attributed to the open pores and partially-closed pores that were penetrated by water through the pore walls and became saturated such that they were damaged in the freeze–thaw cycle [25].

Under the multi-field coupling effect, the thermal conductivity of the rigid polyurethane foam always changed at a faster rate. This was mainly due to the destruction of the pore structure and the escape of gas from the foaming agent, that is, a mixing failure under the four aforementioned weathering resistance tests.

#### 3.1.2. Thermal Conductivity of Foam Concrete

As shown in Figure 5a, in the humid–heat aging test, the thermal conductivity of the foam concrete continuously decreased, and the change rate of the thermal conductivity after 112 days of humidity–heat cycle testing was −5.37%; the possible reason for this was that hydration of the as yet non-hydrated cement in the foam concrete was promoted under the humidity–heat conditions. The formation of hydration products optimized the pore structure of the foam concrete, resulting in a decrease in its thermal conductivity [26].

In the dry–wet cycles, the thermal conductivity of the foam concrete increased from the initial value of 0.0724 W·m^−1^·K^−1^ to 0.0836 W·m^−1^·K^−1^; in the freeze–thaw cycles, the thermal conductivity of the foam concrete rose from the initial value of 0.0789 W·m^−1^·K^−1^ to 0.0980 W·m^−1^·K^−1^. The mass loss increased along with the time in Figure 5b. The main reason for this phenomenon was the destruction of the foam concrete pores, which enhanced the continuity between pores, shortening the heat transfer path, decreasing the thermal resistance of the materials, and thus reducing the thermal insulation performance of the material.

The thermal conductivity of foam concrete increased gradually in high–low temperature cycles, mainly attributing to the foam concrete cracked under these conditions. The number of cracks increased and the effective connectivity of the pores increased. Therefore, it can be concluded that the high–low temperature cycles caused cracking in the foam concrete. Therefore, the path of heat flow in the foam concrete was shortened, the heat resistance was lowered, and the thermal insulation performance of the foam concrete was affected. Under multi-field coupling, the thermal conductivity of the foam concrete increased from 0.072 W·m^−1^·K^−1^ to 0.085 W·m^−1^·K^−1^.

#### 3.1.3. Thermal Conductivity of Vacuum Insulation Panels

It can be seen from Figure 6a that the humidity-heat environment had little effect on the thermal conductivity of the vacuum insulation panels. After 112 days of humid–heat aging, the thermal conductivity increased by 5%, while the mass change was only 0.12% (Figure 6b); after 112 dry–wet cycles, the thermal conductivity remained stable with fluctuations within 0.0002 W·m^−1^·K^−1^. In the high–low temperature cycles, the changes in thermal conductivity of the vacuum insulation panels were not significant, varying from 0.00699 to 0.00709 W·m^−1^·K^−1^. Over all freeze–thaw cycles, the thermal conductivity of the vacuum insulation panels was between 0.007083 and 0.007135 W·m^−1^·K^−1^; meanwhile, under the effect of multi-field coupling, the thermal conductivity plot of the vacuum insulation panels was quasi-horizontal. Therefore, it can be inferred that the vacuum insulation panels provided good weathering resistance because the inner core material of such panels contained micron-sized silica particles. When the surface barrier film was not pierced, all constituent materials of the vacuum insulation plates were covered by a barrier film, which was air-isolating and damp-proof. The inner core material is a chemically stable microsilica powder. Under multi-field coupling with high-temperature and low-temperature alternations, the chemical properties of the material will not change; the plastic deformation ability of the microsilicon powder, the core material of vacuum insulation panels, is improved by vacuum treatment. Vacuum insulation panels can achieve a compressive strength of more than 0.1 MPa without reducing the applied hardness of the vacuum; therefore, the application of a constant stress of 0.5 kN/m^2^ on the top of the panels has a negligible effect. Due to their special structure, vacuum insulation panels can maintain a stable structural performance under the long-term effect of multi-field coupling [27].

#### 3.1.4. Thermal Conductivity of Composite Materials

The performance of composite thermal insulation materials depends on the base materials and is better than the performance of the individual base materials. The composite thermal insulation material is constructed so as to alleviate the adverse effects of the service environment on a single material, and to be synergistic. When the dry–wet cycles are manifested in the service environment, the thermal insulation performance of foam concrete and rigid polyurethane foam will be greatly reduced, which is helpful to maintain the good heat transfer performance of the roofing insulation materials, realizing the same service life of thermal insulation materials and the structure; rigid polyurethane foam materials are not resistant to humid–heat aging, while vacuum insulation panels and foam concrete are less affected in hot, damp environments.

It can be seen from Figure 7a,c, during the high–low temperature cycles, that the thermal conductivity curves of FC + RPUF (foam concrete and rigid polyurethane foam composite) and VIP + FC (vacuum insulation panels and foam concrete composite) were similar. This was because both thermal insulation composite materials used binding mortar as the bonding material. The thermal expansion coefficient of the binding mortar was 12,000 × 10^−6^ mm/(m·K), while that of the foam concrete was 8 × 10^−6^ mm/(m·K), wherein the difference between the two was three orders of magnitude. In the high–low temperature test environment, the materials underwent repeated thermal expansion and contraction. In the high–low temperature cycle test, due to the extreme difference in the coefficients of thermal expansion between the binding mortar and foam concrete, and the low tensile strength of foam concrete, foam concrete in the FC + RPUF and VIP + RPUF samples (Figure 7b) cracked under tension, with cracking evident throughout. These criss-crossed cracks formed a zone of concentrated heat flow, which reduced the thermal insulation performance of the materials. During the test, the number of cracks increased as the number of cycles increased, and the thermal conductivities of the two samples also increased.

#### 3.1.5. Density and Compressive Strength of the FC, RPUF, and VIP

After 112 days of cycling weatherability tests, the density and compressive strength of FC, RPUF, and VIP were measured. It can be seen in Figure 8 and Figure 9 that with the increase of mass loss rate, the density and compressive strength of materials gradually decreased.

### 3.2. Simulation Results and Discussions

Based on the preliminary experimental results, and combined with the humid–heat coupling and heat–solid coupling equations, the combination of 25 mm foam concrete plus 2 mm binding mortar plus 30 mm rigid polyurethane foam (length: 300 mm, width: 300 mm) was selected for the composite insulation material tests. The middle of the bottom (point A (150, 150, 0)) was selected for qualitative numerical simulation calculations, and the results are shown in Figure 10.

In Figure 10, compared with the temperature at point A when only heat transfer was considered, the temperature at point A during heat–solid coupling was higher, but the difference was negligible. It can be deemed that the roof load had no influence on the heat transfer process of the thermal insulation system. It can be concluded that the thermal insulation materials were affected by tensile forces, pressure, shearing forces, and bending moments, and a certain amount of compression in the direction of the force, but the pore structure and the thermal performance of the undamaged materials did not change to any significant extent.

Compared with the temperature at point A under a heat–solid coupling effect or under the sole effect of heat transfer, the temperature at point A under humid–heat coupling (waterproof) effects was lower. The reason was that the water-repellent effect of the combination was good, the liquid water on the roof could not penetrate the materials, and a water layer was formed (the water layer was assumed to be 3 mm deep in the simulation), which indirectly increased the length of the heat transfer path.

Compared with the temperature at point A under a heat–solid coupling effect, or under the sole effect of heat transfer, the temperature at point A under humid–heat coupling (non-watertight conditions) effect was much higher. The reason was that without considering the effect of the waterproof layer, the liquid water penetrated the pores of the materials and the gaps between components. Since the thermal conductivity of liquid water is much higher than that of air, heat transfer between the panels was improved [28]. According to the humid–heat coupling equation, when the humidity increases, the water vapor flux increases, the heat transfer is improved, and the thermal insulation performance of the composite thermal insulation materials declines.

## 4. Analysis of Factors Affecting Heat Transfer

### 4.1. Structural Design

Based on the preliminary experiment results, the initial design scheme of the roofing construction parts is shown in Figure 11a. The inner core of the construction parts consisted of two 20-mm thick vacuum insulation panels that overlapped each other, and the other parts were sprayed and fixed with rigid polyurethane foam. All surfaces of the parts were covered with 0.8 mm stainless steel plates, and the parts at the edge of the roof were surrounded by stainless steel plates. This design can improve the pressure resistance of the roof panels, and furthermore, the panels can function as a waterproof layer. The components were pre-fabricated in a factory and were connected through slots during construction.

Based on the finite element characteristics of COMSOL and the heat–solid coupling equation, the two sides of the construction parts were set as a fixed end and as a thermal insulator respectively; the other parts were free elements. The impact of the structural deformation on the total heat transfer coefficient of the components is shown in Figure 11b. Point A is on the interface between the upper 304 steel plate and the rigid polyurethane foam interface, point B is on the interface between the upper vacuum insulation panel and the rigid polyurethane foam material, point C is on the interface between the lower vacuum insulation panel and the rigid polyurethane foam material, and point D is on the interface between the lower 304 steel plate and the rigid polyurethane foam material.

In Figure 12, compared with the internal temperature of the parts when structural deformation was not considered, the internal temperature of the parts was higher when structural deformation was considered, and the temperature of the bottom part increased by 5%. A possible reason for this was that the coefficients of thermal expansion of the materials in the roofing insulation system were different. When structural deformation was considered, the small cracks inside the materials further accelerated any heat transfer. In the actual construction process, cracks in the roofing components generated by manual handling and erection/installation should be avoided and the complete roofing insulation performance should thus be guaranteed.

In the next step, the stress changes at points A, B, C, and D, and the displacement of point A, were calculated (Figure 13 and Figure 14).

As shown in Figure 13, a large thermal stress was generated at the interfaces between different materials, and the stress on the contact surface between the steel and the rigid polyurethane foam was particularly large at 2.2 × 10^5^ N/m^2^. As shown in Figure 14, the heat conduction process caused a significant expansion deformation of the structure of the roofing thermal insulation components, where the maximum deformation time was 1.5 h and the maximum displacement was 2.05 mm. The deformation increased and then decreased because the outermost layer of the part structure was steel, which has a high thermal conductivity and therefore allowed a rapid rate of heat flux, resulting in the rapid thermal expansion of the steel *ab initio*. Under a steady state heat flux, the temperature of the structure became uniform and the overall expansion was decreased.

### 4.2. Simulation of the Heat Bridge

To analyze the influence of the heat bridge scheme on the heat transfer coefficient, a numerical simulation test of the heat bridge was conducted. Figure 15 shows that the heat bridge effect prevailed at contact points between surrounding steel plates. Two structural forms were tested: in the first, small holes (at 0.3 m spacings) with a diameter of 0.05 m were formed on both sides of the steel plates to reduce the area through which heat could pass; in the second form, a layer of insulating material (1 mm aerogel) was sprayed onto the contact surfaces at both sides of the steel plates to block the connection between the surrounding steel plates and the upper and lower steel plates in the insulation system (Figure 15a,b, respectively).

These two forms of heat bridge were applied in the simulated high summer temperature conditions and compared with the initial scheme. The temperature–time curve for point A (1.5 m, 0.5 m, 0) was calculated and the three schemes were compared (Figure 16).

Figure 16 shows that the holes in the surrounding steel plates had little effect on the overall heat transfer behavior. On the one hand, the small pore size and the insufficient number of holes may be the cause of a large heat transfer zone; on the other hand, the minor change to the heat transfer path during heat transfer may have resulted in a heat transfer state similar to that of the initial scheme. On the contrary, the connection between the upper and lower steel plates and the surrounding steel plates can be effectively cut by attaching a 1-mm-thick aerogel layer to both sides of the steel plates, so that the thermal insulation performance can be significantly improved. The reason for this is that the thermal conductivity, density, and specific heat capacity of thermal insulation materials, such as aerogels, are very low. In the heat transfer process, it can be considered that the surrounding parts of the roof system is thermally insulated, such that the heat transfer path is theoretically from top to bottom [29,30]. Therefore, it is necessary to select a suitable roofing structure for the design of the heat bridge according to the prevailing environmental conditions and local cost constraints.

## 5. Conclusions

Weathering resistance tests on three types of typical insulation materials were conducted, and the evolution of the macroscopic properties (thermal conductivity and mass loss rate) of three thermal insulation materials was obtained. Various schemes combining two types of thermal insulation materials were proposed to achieve energy-saving effects. In the study, the influences of environmental factors (load, temperature, and humidity) were also considered via numerical simulations using COMSOL software. The main conclusions drawn from this study are summarized as follows:(1)In the weather resistance test, the thermal conductivity of the rigid polyurethane foam board and foam concrete slab increased gradually with time, while the vacuum insulation board had excellent weatherability under the condition that the surface barrier film was not pierced. The weatherability of the composite materials combined the advantages of various materials and had a better thermal insulation performance.(2)The load had little effect on the heat transfer process of the roof system. When the roof had a relative humidity and the waterproof layer was not destroyed, the relative humidity slowed down the heat transfer process. When the role of the waterproof layer was not considered, the relative humidity accelerated the heat transfer process, which had 1.11-fold increase over the waterproof roof.(3)In the process of heat conduction through the roof insulation material parts, due to the uneven temperature, the 304 steel plate produced greater structural self-stress. The stress representative value at the interface between 304 steel plate and rigid polyurethane foam was 220 kPa. The maximum structural displacement of the parts was 2.05 mm due to uneven heating. Due to structural deformation, the heat conduction process occurred. In the middle, the bottom temperature of the product increased by 5%.(4)Through the simulations, it was found that the spraying aerogel bonding method could effectively play the role of a heat-breaking bridge. The thermal insulation performance of the roof increased 2 times than the area of the roof.

## Figures and Tables

**Figure 1 materials-12-03348-f001:**
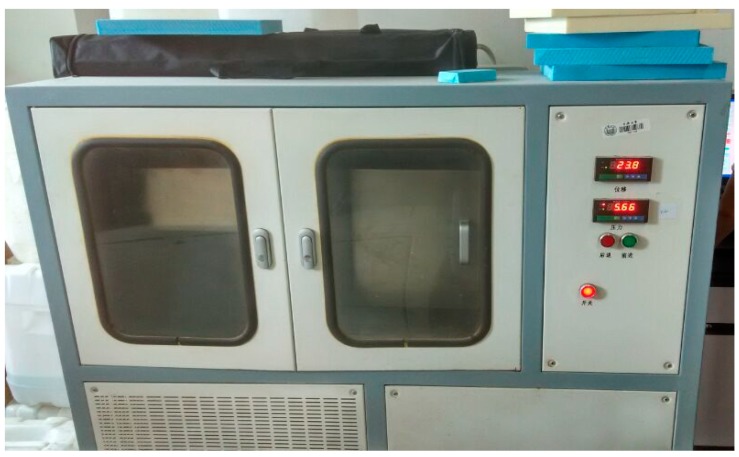
Test instrument.

**Figure 2 materials-12-03348-f002:**
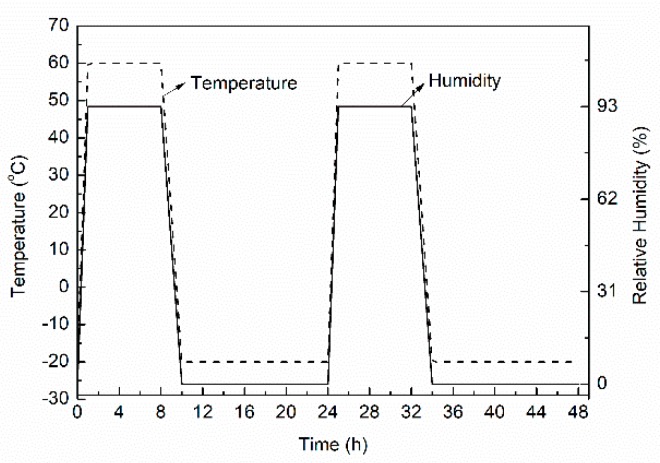
Schematic diagram of multi-field temperature–humidity cycling tests.

**Figure 3 materials-12-03348-f003:**
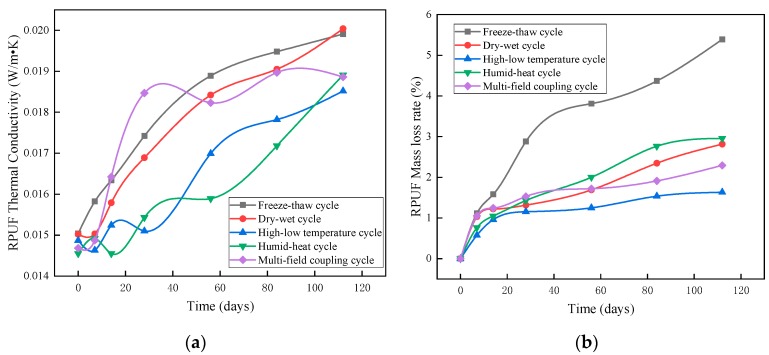
Changes in the rigid polyurethane foam (RPUF) in weathering resistance tests: (**a**) change in thermal conductivity of the rigid polyurethane foam, and (**b**) change in mass loss of rigid polyurethane foam.

**Figure 4 materials-12-03348-f004:**
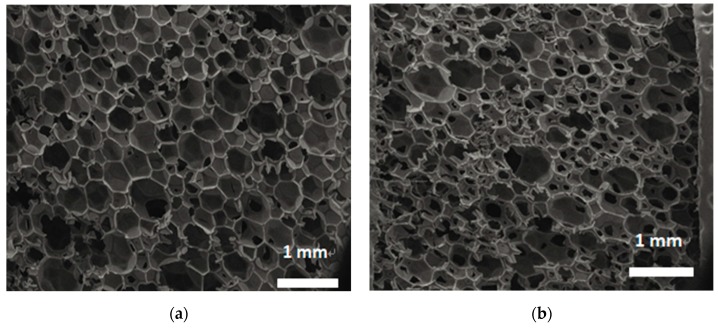
Changes in the rigid polyurethane foam from dry–humidity cycling: (**a**) initial sample, and (**b**) after 112 days of dry–humidity cycling.

**Figure 5 materials-12-03348-f005:**
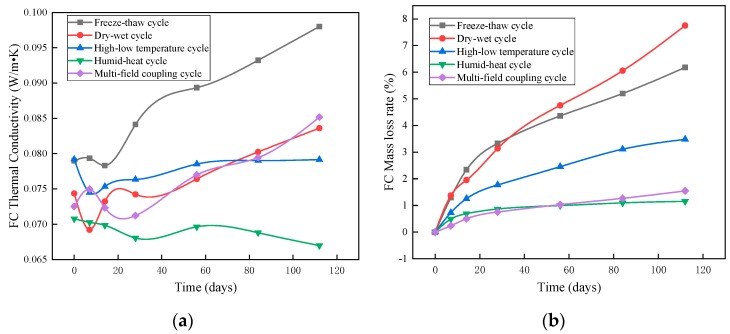
Changes of foam concrete in weathering resistance tests: (**a**) thermal conductivity of foam concrete, and (**b**) mass loss of foam concrete.

**Figure 6 materials-12-03348-f006:**
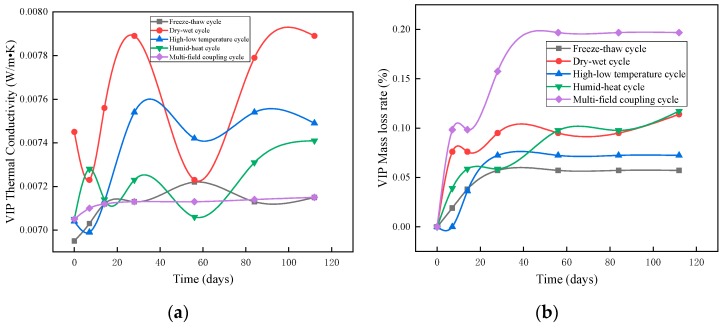
Changes in vacuum insulation panels in weathering resistance tests: (**a**) changes in thermal conductivity of vacuum insulation panels, and (**b**) changes in mass loss of vacuum insulation panels.

**Figure 7 materials-12-03348-f007:**
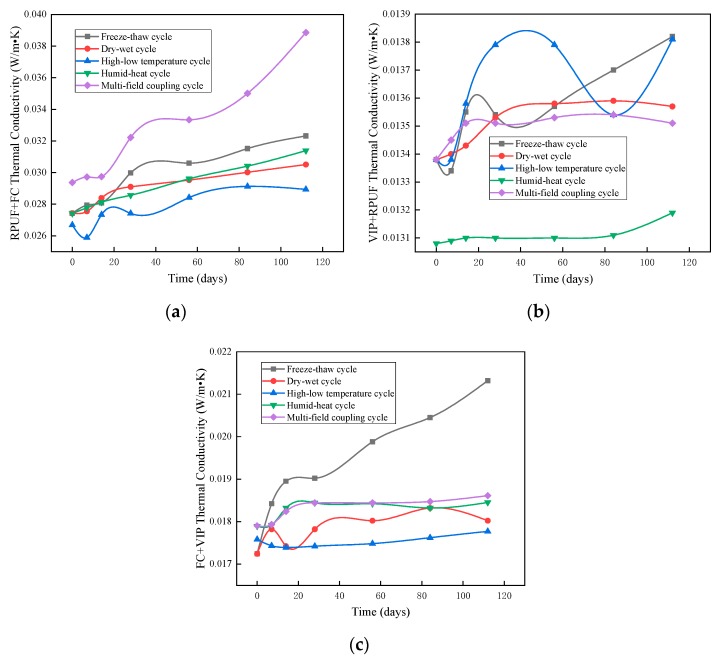
Changes in thermal conductivity of composite materials in weathering resistance tests: (**a**) foam concrete (FC) and rigid polyurethane foam composite (RPUF), (**b**) vacuum insulation panels (VIP) and RPUF, and (**c**) VIP and FC.

**Figure 8 materials-12-03348-f008:**
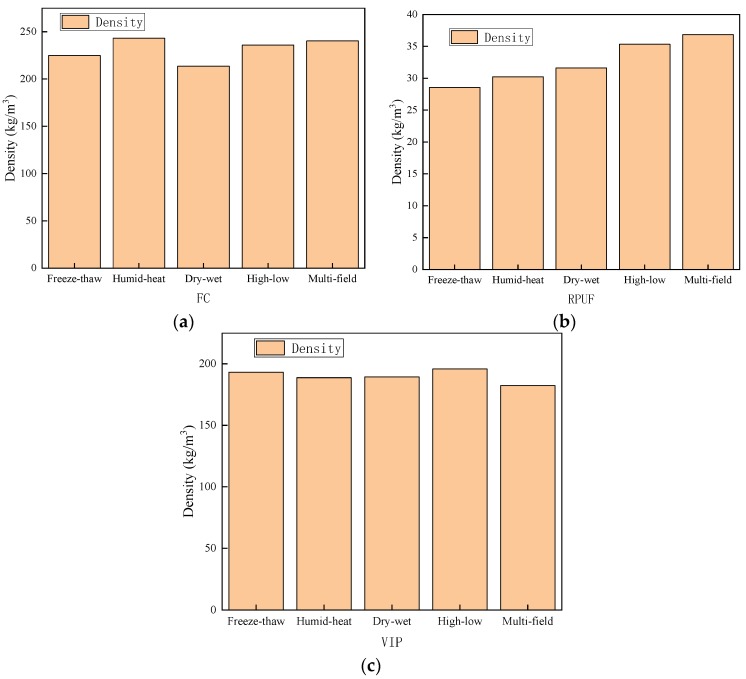
Changes in density of materials after the 112 days of cycling: (**a**) FC, (**b**) RPUF, and (**c**) VIP.

**Figure 9 materials-12-03348-f009:**
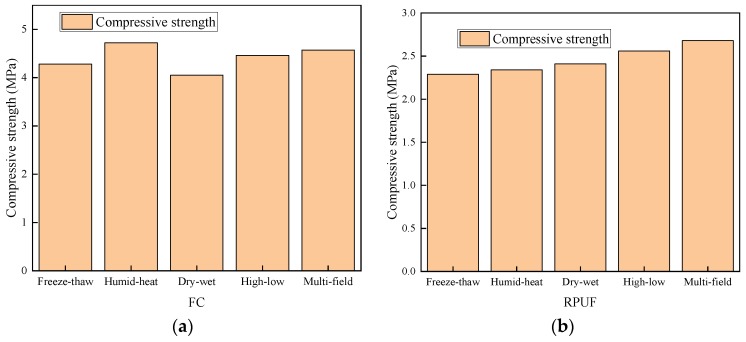
Changes in compressive strength of materials over the 112 days of cycling. (**a**) FC, (**b**) RPUF, and (**c**) VIP.

**Figure 10 materials-12-03348-f010:**
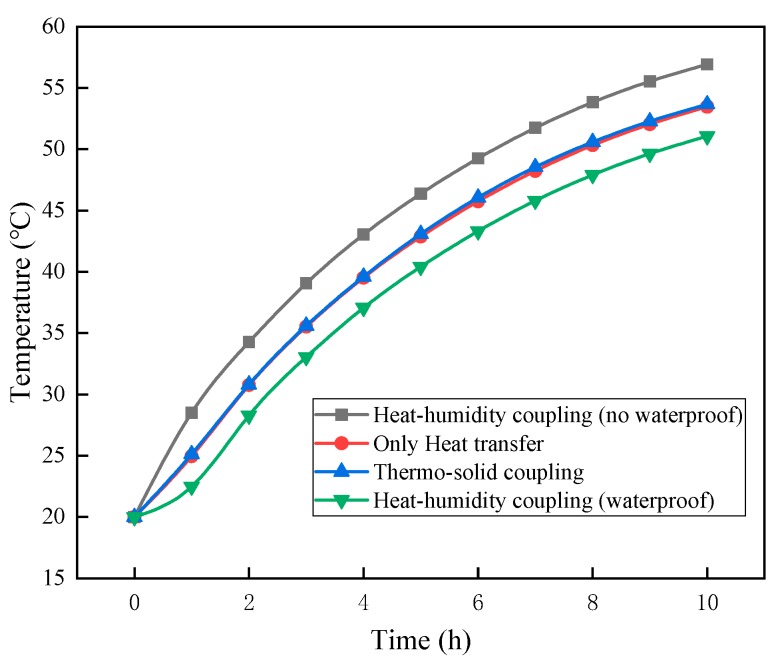
Indoor temperature curve under different coupling effects.

**Figure 11 materials-12-03348-f011:**
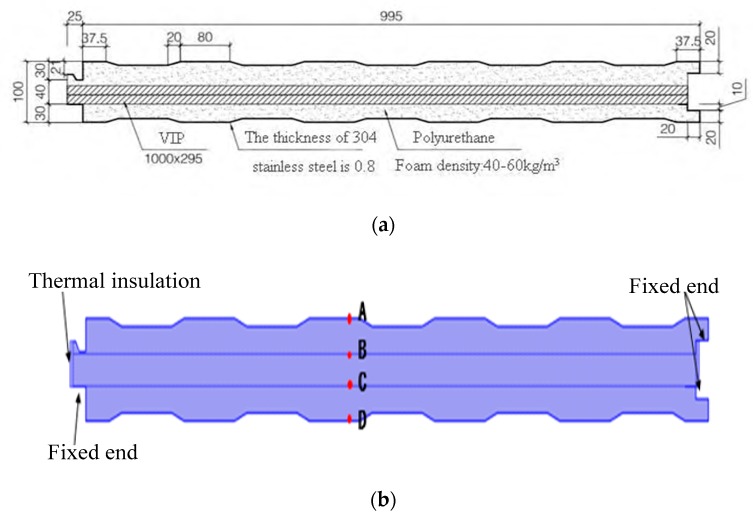
Diagram of sample component parts: (**a**) design scheme, and (**b**) simulation layout.

**Figure 12 materials-12-03348-f012:**
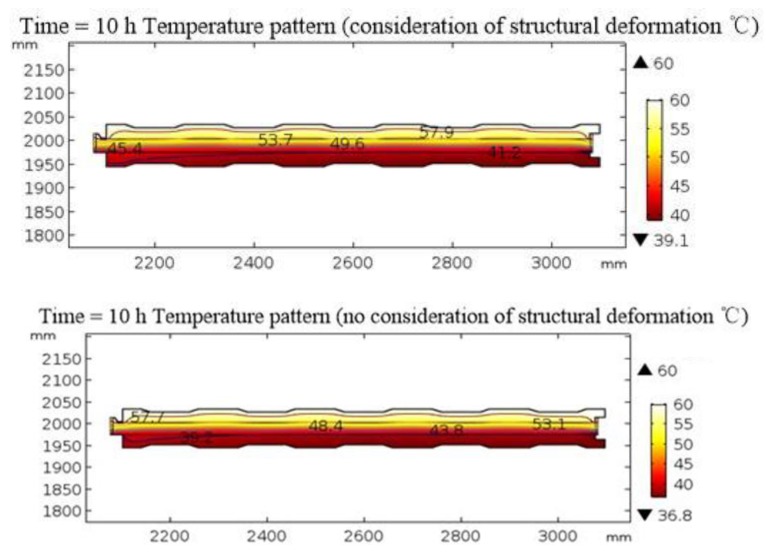
Effect of structural deformation on heat transfer.

**Figure 13 materials-12-03348-f013:**
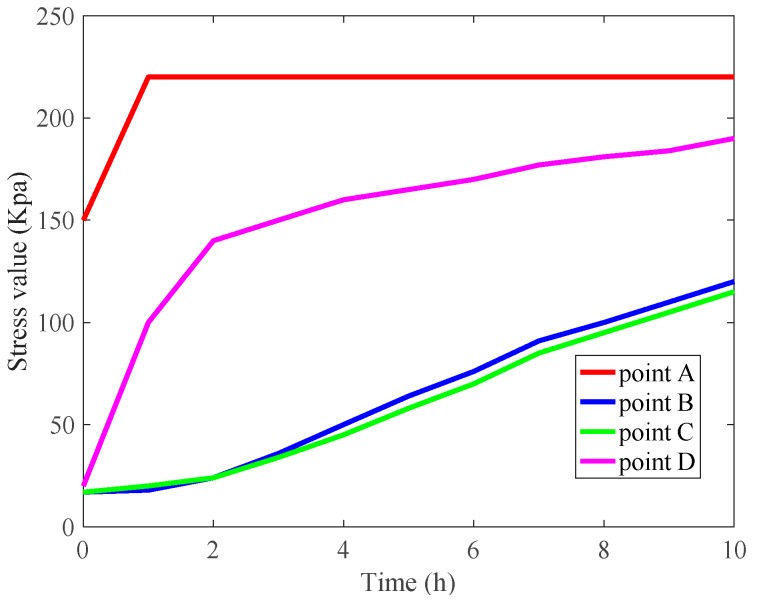
Stress change at the interfaces between components.

**Figure 14 materials-12-03348-f014:**
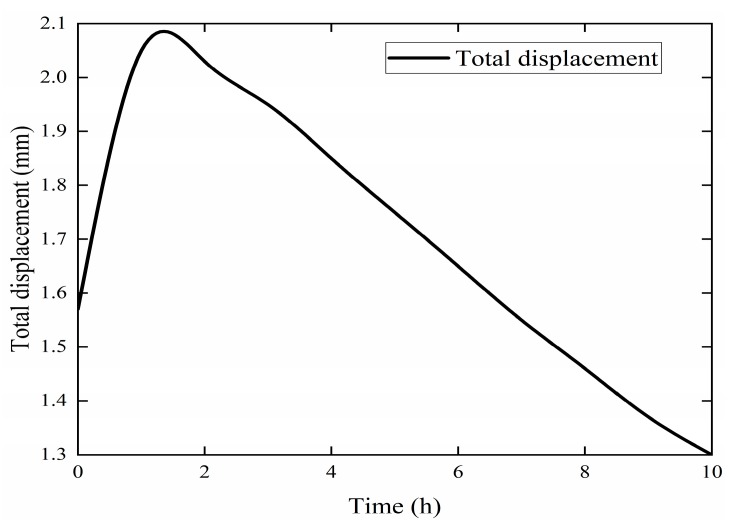
Displacement at point A.

**Figure 15 materials-12-03348-f015:**
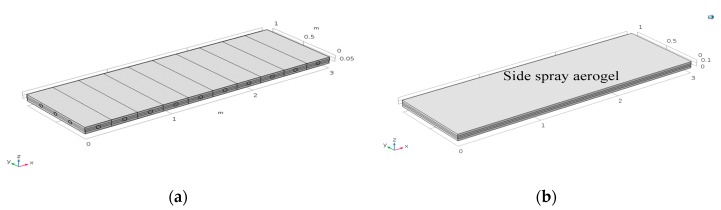
The two forms of heat bridge: (**a**) construction form, and (**b**) aerogel bonding.

**Figure 16 materials-12-03348-f016:**
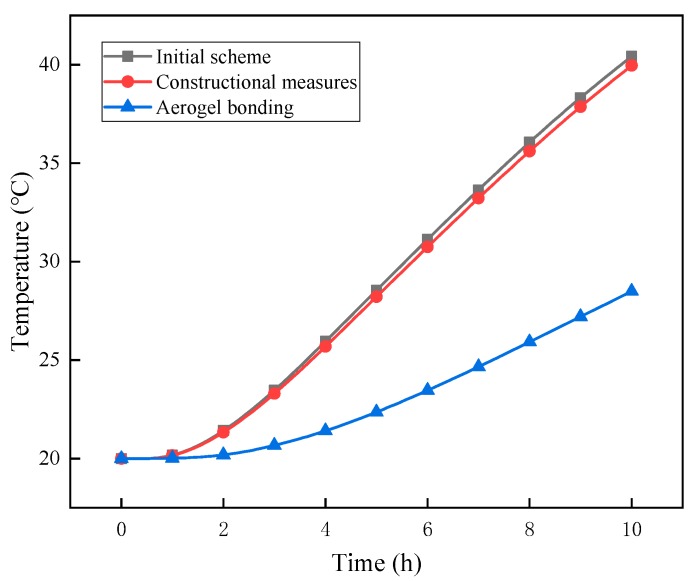
Temperature–time curves at point A for different forms of heat bridge.

**Table 1 materials-12-03348-t001:** Materials parameters.

Materials	Heat Capacity (J/kg·K)	Density (kg/m^3^)	Blowing Agent
FC	1050	247	dicyandiamide
RPUF	1380	39.15	hydrocarbon
VIP	1280	196.48	/

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
