# Peer review of "Weathering of Roofing Insulation Materials under Multi-Field Coupling Conditions"

_materials, 2019, doi:10.3390/ma12203348_

Round 1
Reviewer 1 Report
Thank you for the opportunity to review the article entitled Weathering of Roofing Insulation Materials under Multi-field Coupling Conditions by Zhou Shuangxi, Ding Yang, Wang Zhongping, Dong Jingliang and Li Ruguang. The article is very interesting. It covers a very important subject matter, which is thermal insulation materials used in civil engineering. However, there are some ambiguities in which the authors must respond and add some information. The article also requires minor editorial changes in accordance with the Instructions for authors. My comments, remarks and suggestions are presented below.
MY COMMENT, REMARKS AND SUGGESTIONS
1) Authors must change the way of citing. The reference should not provide in the superscript.
2) Section 2.1 Test Materials is not described in detail. Authors should better characterize the used materials, because this has a significant impact on the interpretation of the obtained results. For example, in the case of polyurethane, it was not clearly indicated: what type it was (only short information in result and discussion section, that was a rigid foam), what was the blowing agent, what was its apparent density? These parameters affect the thermal conductivity of this material. Also the characteristics of other materials must be added. Lack of this information makes it impossible to do again this studies in others research centers.
3) The authors use the abbreviation "d" for the word "day". This abbreviation is rarely used. I suggest changing "... 7 d, 14 d, 28 d, 56 d, and 112 d" to "... 7, 14, 28, 56 and 112 days" throughout manuscript text.
4) Line 110 – is „7h”, should be „7 h”.
5) Section 2.3 The authors did not provide any information about software producer.
6) Sections 2.3.1 and 2.3.2 - I suggest to write the units in round brackets, because in present form the readability of these sentences is difficult.
7) Line 162 – wrong unit for specific heat and no unit for Q.
8) General remark to the entire text. The authors write "polyurethane" throughout the text. This word can be confusing for potential reader, because there are many types of polyurethanes. Several of them are used in civil engineering. Authors used in their research the rigid polyurethane foams (or polyisocyanurate foams, I cannot confirm that because the description of the materials is not detail). Therefore, the word 'polyurethane' should be changed to 'polyurethane foam' or 'rigid polyurethane foam' throughout the text. Optionally, PUF or RPUF can be used.
9) Section 3.1.1 - How was thermal conductivity of PU foam measured? Only by mathematical calculations or by using the Heat Flow Meter apparatus? Lack of information. In my opinion, measuring with an apparatus would be more reliable than simulation.
10) The authors wrote "... with increased aging time, the polyurethane pore structure was destroyed". Did the authors confirm this by SEM or other structure analysis?
11) „It can be inferred that the diffusion coefficient of the foaming agent gas in the closed pores of the matrix increased in this high-temperature environment [20].” This statement should be developed taking into account the specific blowing agent that was used to obtain a RPUF. The λ coefficient depends strictly on the type of blowing agent.
12) „Due to the high closed pore ratio of the rigid polyurethane foam…”. What was the high content of closed cells? This parameter is very important during analysis of thermal conductivity of RPUFs. Authors should check how this parameter changed during cycles, because it is responsible for the increase of the λ In addition, it is difficult to determine whether changes in this factor are high or not, because there has not information about the used foam in the Test Materials section.
13) Figure 4 a. Values of changes in the range below 0.001 W/(m*K) are within the error of measurement.
14) Figure 5 b. Values of changes in the range below 0.001 W/(m*K) are within the error of measurement.
15) The authors use two types of abbreviations F-P, V-F, V-P and FC + PUR, VIP + FC, VIP + PUR. They have to choose one and use it in the text.
16) The Conclusion section must be reworded. The authors must add the most important numerical values of the obtained test results.
17) Authors should add DOI numbers to cited references.
Author Response
Dear Editor,
Thanks for your kind letter. Our manuscript has been improved based on the reviewer’ thoughtful and detailed comments, and hence we are submitting the revised manuscript. All comments made by the reviewers are addressed, and the substance of many of the comments has been incorporated in the revised manuscript. Detailed responses to the comments of the reviewers are listed in the attached files. Additionally, a thorough check on the revised manuscript has been carried out to minimized typographical, grammatical, and bibliographical errors. All the changes and improvements in the revised manuscript are marked with red. We hope that these changes lead to the acceptance of the manuscript, and look forward to hearing from you.Best wishes for you!

Reviewer 2 Report
General Comments: The research of Shuangxi et al. deals with the weathering of roofing insulation materials under multi-field coupling conditions. This is an important contribution to the field of insulation materials. However, I feel that it needs some major work to bring it up to a suitable standard for publication before to be accepted. Thus, revisions must be made to be considered in this journal. I provide some detailed objections and comments to support my revisions:
1.Please update the literature review. Just to mention a few examples:
Sustainable thermal insulation biocomposites from rice husk, wheat husk, wood fibers and textile waste fibers: Elaboration and performances evaluation, Industrial Crops and Products, 2019 135, 238-245
Mechanical and thermal insulation properties of elium acrylic resin/cellulose nanofiber based composite aerogels, Nano-Structures & Nano-Objects, 2017 12, 68-76
Applications of polysaccharide and protein based aerogels in thermal insulation, Biobased Aerogels: Polysaccharide and Protein-Based Materials, 2018, 261-294
2.How the density of the samples affected after exposure to different environmental conditions. The thermal conductivity of the samples should be correlated with the density of the samples.
3. Authors have reported the mass loss of the samples under different conditions, what could be the material leaching from the samples during the conditioning.
4. In order to explain the changes in the thermal conductivity of the samples under different environment, authors should perform morphological analysis of the samples before and after conditioning at different environmental conditions.
5. In addition to thermal conductivity, what is the effect of mechanical properties after conditioning of the samples at different environmental conditions.
Author Response

(The authors gave the same response as above.)

Round 2
Reviewer 1 Report
The authors of the article entitled "Weathering of roofing insulation materials under multi-field coupling conditions" significantly improved the submitted manuscript. I think that it is suitable for publication in the journal Materials in present form.
Reviewer 2 Report
The authors have addressed all the comments, so now I am recommending this manuscript to publish in this journal.